# Factors associated with the age of the onset of diabetes in women aged 50 years or more: a population-based study

Ana L R Valadares,[1] Vanessa S S Machado,[1] Lúcia S Costa-Paiva,[1] Maria H de Sousa,[2] Aarão M Pinto-Neto[1]

[1]Department of Obstetrics and Gynecology, School of Medical Sciences, State University of Campinas (UNICAMP), Campinas, Brazil
[2]Department of Statistics, Campinas Center for Research in Reproductive Health (CEMICAMP), UNICAMP, Campinas, Brazil

**Correspondence to**
Dr Ana Lúcia Ribeiro Valadares; anarvaladares@gmail.com

## ABSTRACT

**Objective:** Investigate factors associated with the onset of diabetes in women aged more than 49 years.

**Design and methods:** Cross-sectional, population-based study using self-reports with 622 women. The dependent variable was the age of occurrence of diabetes using the life table method. Cox multiple regression models were adjusted to analyse the onset of diabetes according to predictor variables. Sociodemographic, clinical and behavioural factors were evaluated.

**Results:** Of the 622 women interviewed, 22.7% had diabetes. The mean age at onset was 56 years. The factors associated with the age of occurrence of diabetes were self-rated health (very good, good) (coefficient=−0.792; SE of the coefficient=0.215; p=0.0001), more than two individuals living in the household (coefficient=0.656, SE of the coefficient=0.223; p=0.003), and body mass index (BMI) (kg/m$^2$) at 20–30 years of age (coefficient= 0.056, SE of the coefficient=0.023; p=0.014).

**Conclusions:** Self-rated health considered good or very good was associated with a higher rate of survival without diabetes. Sharing a home with two or more other people and a weight increase at 20–30 years of age was associated with the onset of type 2 diabetes.

## Strengths and limitations of this study

- Knowledge based on an epidemiological population study about factors associated with the onset of diabetes in women aged 50 years or more.
- Information that body mass index (BMI) control in young adults, better self-rated health and improvement of socioeconomic conditions could prevent the onset of diabetes in the ageing process.
- Observation that there was no association between menopausal status and the onset of diabetes.
- The age of the occurrence of diabetes was also based on the report of the diagnosis made by the physician and other degrees of abnormal glucose tolerance were not taken into account.
- It was not possible to consider all factors that can impact on risks related to diabetes like health problems, gestational diabetes and dietary intake or type of foods consumed.

## INTRODUCTION

Type 2 diabetes is a chronic, heterogeneous, progressive metabolic disease that is characterised by insulin resistance. The relevance of this condition lies in its high prevalence and incidence, the individual burden of disease in patients due to macrovascular and microvascular complications, and the associated costs to the healthcare system.[1] In women with diabetes, life expectancy was found to be 5.8 years shorter than in women without diabetes, irrespective of income.[2]

The prevalence of diabetes has increased worldwide, reaching epidemic proportions in recent years, as a result of the ageing population and obesity.[3–5] It is estimated that in 2011, 8.6% of individuals in Central and South America had diabetes, and predictions suggest that this percentage will reach 10.1% by 2030. In Brazil, the prevalence of diabetes is 13.5% in individuals aged 30–79 years[6] and 18.7% in women aged above 60 years.[7] Hospitalisations due to diabetes mellitus account for 9% of hospital spending within the Brazilian National Health System (*Sistema Único de Saúde*—SUS).[8] In general, the prevalence of diabetes increases with ageing, particularly after 45–50 years of age, and the presence of diabetes also constitutes an important risk factor for cardiovascular disease. Physical inactivity, an inadequate diet and an increase in the prevalence of obesity are factors held responsible for the global expansion of diabetes.[5]

One of the most used methods for estimating the prevalence of common chronic diseases such as type 2 diabetes is a nationwide or regional population survey. Such surveys are usually restricted to self-reported data on diabetes; however, specificity has been found to be high, with the data from these surveys correlating well with the actual occurrence of the disease.[9][10] Diabetes is at the centre of behavioural problems, and psychological and social factors play a crucial role in its management;[11] therefore, it is important to know which factors contribute towards its onset.

The objective of this study was to investigate the factors most strongly associated with age at onset of diabetes in women aged 50 years or more in a population-based study conducted in Brazil.

## METHODS AND PROCEDURES
### Subjects
A cross-sectional, population-based study using data derived from self-reports was conducted between 10 May and 31 October 2011 in the city of Campinas, São Paulo, Brazil. Sixty-eight census sectors (the primary sampling units) of the city of Campinas, Brazil were randomly selected by simple random sampling or equal probability of selection. The selection procedure was performed according to a table of random numbers generated from a list supplied by the Brazilian Institute of Geography and Statistics (IBGE) and classified according to the sector identification number (ID__). Prior to selection, the number of women aged 50 years or more living in each census sector (women eligible for the study) was determined. Sectors with fewer than 10 women in this age group were grouped together with the neighbouring sector holding the subsequent ID number. Research assistants, guided by maps of each census area, went to the odd-numbered houses and verified whether there were any women aged 50 years or more living there. If there were eligible women residing at the address, they were invited to participate in the research project, and if they agreed, a questionnaire was applied personally by interviewers trained at the Campinas Center for Research and the Control of Maternal and Child Diseases (CEMICAMP) until 10 eligible women had been interviewed in each sector. If it proved impossible to interview 10 women in any given sector using this methodology, work was then resumed in that sector by visiting the addresses not included at the first attempt (ie, the even-numbered houses). A total of 622 women effectively participated in this study, since 99 of 721 invitations (13.7%) were declined.

### Sample size
The target population consisted of all the female residents of Campinas, São Paulo, Brazil, who were aged 50 years or more in 2007. This made a total of 131 800 women. To calculate sample size, the prevalence of diabetes was estimated at 13.5%.[6] A type I error of 5% was defined, with a margin of error of 4% (the absolute difference between the proportion in the sample and that of the population), resulting in a sample size of 280 women. Taking into consideration a possible loss of 10% of the participants, the minimum sample size was increased to 308 women. The final sample obtained consisted of 617 women aged 50 years or more.

This study forms part of a larger project conducted to evaluate the health conditions of women aged 50 years or more. The project was approved by the internal review board of CAISM/UNICAMP and was conducted in compliance with the current version of the Declaration of Helsinki and with Resolution 196/96 of the Brazilian National Committee for Ethics in Research (CONEP) and its subsequent revisions.

### Inclusion and exclusion criteria
Women aged 50 years or more were eligible, while those with any factor that prevented the interview from taking place were excluded. Precluding factors included lack of cognitive ability to answer the questionnaire, prior commitments and incompatibility of schedules.

### Instrument
The participants answered a structured, pre-tested questionnaire created on the basis of three pre-existing questionnaires. Of these, two were Brazilian questionnaires, one of which was part of the SABE project on health, well-being and ageing in Latin America and the Caribbean,[7] while the other formed part of a population-based survey denominated VIGITEL 2008, conducted by the Brazilian Ministry of Health.[12] The third questionnaire was used in the 'Women's Health and Aging Study', a nationwide study conducted in the USA.[13] The present questionnaire was divided into five sections: sociodemographic evaluation, health-related habits, self-perception of health, and evaluation of functional capacity and health-related problems.

### Variables
The independent variables consisted of: age (in years), marital status, years of schooling, number of people living in the household, skin colour, smoking and alcohol consumption, having private medical insurance, practice of physical exercise, having stopped menstruating more than a year ago; physician's diagnosis of menopause, body mass index (BMI) at 20–30 years of age, current BMI, and self-perception of health.

The dependent variable was age at onset of diabetes reported by women at the time of the interview. This information was obtained by asking women if they had the disease and whether it was diagnosed by a physician. With a positive answer, the individual was then asked about the time since diagnosis and on treatment. Thus, the presence of diabetes was further validated.

### Data analysis
First, the age of onset of diabetes in annual intervals, reported by women at the time of the interview, was

used to calculate the cumulative continuation rates (survival) without diabetes, using the life table method. If the woman had not experienced diabetes at the time of the interview, it was considered censored data.[14] Next, Cox multiple regression models were adjusted to analyse the rates of the age of the onset of diabetes in accordance with various predictive variables: schooling (≤8, >8 years); marital status (with a partner, no partner); skin colour (white, other); number of people living in the household (≤ 2, >2); smoking (never smoked, past or current smoker); number of cigarettes smoked per day currently or in the past (≤15, >15); alcohol consumption (yes, no); frequency of alcohol consumption (none or <12 day/month, other); having private medical insurance (yes, no); weekly practice of physical exercise (yes, no); frequency of physical exercise (≤ 2, ≥ 3 days/week); BMI (kg/m$^2$) at 20–30 years of age; self-rated health (very good, good; other); menopausal (yes, no).

All the women participated voluntarily in the study and signed an informed consent form. The study protocol was approved by the internal review board of the School of Medical Sciences, University of Campinas.

## RESULTS

The sociodemographic characteristics of the women in the study sample are shown in table 1.

Of the 617 women interviewed, 22.7% reported having diabetes. Of the women with diabetes (n=140), the mean age at onset of the disease was 56±11.2 years (median 55 years), reported at the time of the interview (figure 1). The factors associated with the age of occurrence of diabetes were self-rated health (very good, good) (coefficient=−0.792, SE of the coefficient=0.215; p=0.001), more than two people living in the household (coefficient=0.656; SE of the coefficient=0.223; p=0.003); and BMI (kg/m$^2$) at 20–30 years of age (coefficient=0.056, SE of the coefficient=0.023; p=0.014) (table 2). No association was found between menopausal status and diabetes.

## DISCUSSION

The objective of this population-based study was to evaluate factors associated with age at onset of diabetes in women above 49 years.

In the current study, the prevalence of self-reported diabetes was 22.7%, which could lead to misreporting. This finding is consistent with that of other studies. In Brazil, Lebrão et al[7] showed an 18.7% prevalence of self-reported diabetes among women aged above 60 years, and in the USA, for the period 2005–2008, it was estimated that 26.9% of people aged 65 years or more had diabetes, based on both fasting glucose and glycated haemoglobin levels.[15]

Self-rated health considered good or very good was associated with a higher rate of survival without diabetes. In line with this finding, a prospective case–cohort study reported that poor self-rated health was associated with a

higher risk of type 2 diabetes.[16] Previous studies conducted with occupational cohorts have suggested that self-rated health principally indicates physical and mental health problems and, to a lesser extent, age, early life factors, family history, sociodemographic variables, psychosocial factors and health-related behaviour.[17 18] As it was a cross-sectional study, one can only say that there was an association of onset of diabetes with self-rated health. So it was not possible to demonstrate that poor self-rated health was a causative factor or the effect of the onset of diabetes, due to the design of this study.

Sharing a home with more than one person was associated with the presence of diabetes. Reports in the literature on the number of individuals sharing a home and the presence of diabetes are conflicting. In a population cohort that included both men and women, an association was found between living alone and type 2 diabetes in men; however, there was no increased risk for women living alone.[19] Nevertheless, a Swedish study investigating the role of household conditions in the progression from impaired glucose tolerance to diabetes in 461 women aged 50–64 years found that women living alone had a 2.7-fold increased risk of type 2 diabetes even after adjustment for biological risk factors.[20] In other countries, living alone is believed to be related to poor perceived social support, lack of a close confidant and poor emotional support, and may be a proxy for poor social support and consequently social isolation.[21] We may hypothesise that the difference between the findings of this study and those of Lidfeldt et al[20] may be explained by the fact that in Brazil the women most likely to have type 2 diabetes are older and share a home with other people because they require care. In addition, one may also hypothesise that these women may have lower incomes and poorer health conditions. A large body of evidence suggests that socioeconomically disadvantaged groups are at increased risk of type 2 diabetes.[22 23]

A BMI increase at 20–30 years of age was another factor associated with the onset of diabetes. Studies showed that being obese or overweight at a younger age may increase the risk of developing diabetes.[24 25] In a longitudinal study enrolling adults aged above 35 years with no cardiovascular disease or diabetes, which was conducted during a 7-year follow-up period, the BMI cut-off of 30 kg/m$^2$ was associated with a 1.94-fold (1.42–2.66) increased risk of type 2 diabetes.[24] Jeffreys et al[25] Have also demonstrated that overweight at any point in a person's life is associated with an increased risk of developing diabetes and that the risk associated with being overweight is cumulative across the life course.

No association was found between menopausal status and the onset of diabetes in this study. In agreement with this finding, Kim et al[26] reported that in the women in the Diabetes Prevention Program who were at a high risk for diabetes, natural menopause did not increase this risk. Nevertheless, most of the previous studies were

**Table 1** Percentage of women without and with diabetes according to their sociodemographic and behavioural characteristics—bivariate analysis

| Variable | N | Without diabetes (%) | With diabetes (%) | p Value* |
|---|---|---|---|---|
| Age (years) | 617 | | | 0.04 |
| 50–59 | | 41.7 | 30.0 | |
| 60–69 | | 30.8 | 36.4 | |
| ≥70 | | 25.5 | 33.6 | |
| Schooling (years) | 617 | | | 0.20 |
| ≤8 | | 67.9 | 78.6 | |
| >8 | | 32.1 | 21.4 | |
| Marital status† | 609 | | | 0.59 |
| Partner | | 48.7 | 45.7 | |
| No partner | | 51.3 | 54.3 | |
| Skin colour/ethnicity† | 609 | | | 0.76 |
| Other | | 29.0 | 30.7 | |
| White | | 71.0 | 69.3 | |
| Monthly income† | 397 | | | 0.07 |
| ≤US$750 | | 50.8 | 62.2 | |
| >US$750 | | 49.2 | 37.8 | |
| Smoking | 617 | | | 0.22 |
| Never smoked | | 62.5 | 68.5 | |
| Used to smoke | | 37.5 | 31.4 | |
| Smoker | | | | |
| Number of cigarettes smoked per day currently or in the past | 601 | | | 0.92 |
| ≤15 | | 85.4 | 84.6 | |
| >15 | | 14.6 | 15.4 | |
| Alcohol consumption | 617 | | | 0.97 |
| No | | 85.1 | 85.0 | |
| Yes | | 14.9 | 15.0 | |
| Frequency of alcohol consumption | 617 | | | 0.92 |
| None or <1 day/month | | 88.9 | 88.6 | |
| >1 day/month | | 11.1 | 11.4 | |
| Woman has private medical insurance | 617 | | | 0.17 |
| No | | 50.1 | 57.1 | |
| Yes | | 49.9 | 42.9 | |
| Practises physical exercise weekly | 617 | | | 0.16 |
| No | | 62.3 | 69.3 | |
| Yes | | 37.7 | 30.7 | |
| Frequency of physical exercise† | 616 | | | 0.74 |
| Up to 2 days/week | | 73.1 | 75.0 | |
| ≥3 days/week | | 26.9 | 25.0 | |
| BMI (kg/m$^2$)† | 497 | | | 0.007 |
| <25 | | 40.3 | 25.5 | |
| 25–29.9 | | 36.7 | 40.0 | |
| ≥30 | | 23.0 | 34.5 | |
| BMI (kg/m$^2$) at 20–30 years of age† | 440 | | | 0.001 |
| <25 | | 37.2 | 32.2 | |
| 25–29.9 | | 53.3 | 44.1 | |
| ≥30 | | 9.5 | 23.7 | |
| Self-rated health | 617 | | | 0.0001 |
| Other | | 36.7 | 56.4 | |
| Very good, good | | 63.3 | 43.6 | |
| Woman stopped menstruating over 1 year ago† | 615 | | | 0.95 |
| No | | 13.4 | 13.7 | |
| Yes | | 85.6 | 85.3 | |

*$\chi^2$ Test with Yates' correction.
†Some values may not add up to 617 because of missing data.
BMI, body mass index.

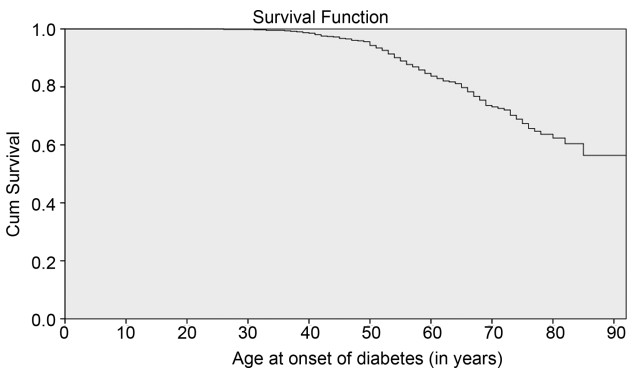

**Figure 1** Age at the onset of diabetes over a lifetime (years). Cumulative survival N=617. Mean age at onset of the disease was 56±11.2 years (median 55 years). Cumulative continuation rate without diabetes was 56% at 92 years of age.

small, hence insufficiently powered to answer this question.[27] One also has to consider that the absence of association between menopausal status and risk for diabetes may be due to the majority of women being already postmenopausal at the time of onset of diabetes in this study.

Some limitations of this study must be taken into consideration. As in most population-based studies, the presence of diabetes mellitus was determined on the basis of self-reported physician-diagnosed diabetes, and confirmation of this diagnosis was not made. Nevertheless, the onset of a disease so important like diabetes is generally remembered, which decreases the risk of remembering bias.[28] It was also not possible to consider all of the factors that can impact the risk of the onset of diabetes like health problems, gestational diabetes and dietary intake or type of foods consumed.[29]

**Table 2** Variables associated with the presence of diabetes—Cox multiple regression analysis (n=428)

| Variable | Coefficient | SE of the coefficient | p Value |
|---|---|---|---|
| Self-rated health (very good, good) | −0.792 | 0.215 | <0.001 |
| Number of individuals living in the household (>2) | 0.656 | 0.223 | 0.003 |
| BMI (kg/m$^2$) at 20–30 years of age | 0.056 | 0.023 | 0.014 |

Predictor variables taken into consideration: schooling (≤8 years: 0; >8 years: 1); marital status (with a partner: 1; no partner: 0); skin colour (white: 1; other: 0); number of people living in the household (≤2: 0; >2: 1); smoking (never smoked: 0; past or current smoker: 1); number of cigarettes smoked per day currently or in the past (≤15: 0; >15: 1); alcohol consumption (yes: 1; no: 0); frequency of alcohol consumption (none or <1 day/month: 0; other: 1); woman has private medical insurance (yes: 1; no: 0); practises physical exercise weekly (yes: 1; no: 0); frequency of physical exercise (≤2 days/week: 0; ≥3 days/week: 1); BMI (kg/m$^2$) at 20–30 years of age; self-rated health (very good, good: 1; other: 0); menopausal (yes: 1; no: 0).
BMI, body mass index.

Furthermore, in this study, the age of the occurrence of diabetes was also based on the report of the diagnosis made by the physician and other degrees of abnormal glucose tolerance were not taken into account. The reliability of self-reported diabetes mellitus has been previously validated.[2] The fact of the study having a population-based nature represents an important strongpoint. The representativeness of the population sample allows these conclusions to be extrapolated to the entire population of women aged 50 years or more in a Brazilian city. Population-based estimates of the age of occurrence of diabetes in women aged 50 years or more and its associated factors are important for understanding this issue in women's lives as they age, while designing interventions in the field of diabetes prevention requires good knowledge of region-specific trends.

## Conclusions

Self-rated health considered good or very good was associated with a higher rate of survival without diabetes. Sharing a home with two or more other people and a weight increase at 20–30 years of age was associated with the onset of type 2 diabetes.

These results contribute to highlighting the need to target weight control interventions earlier in life and for measures aimed to improve women's socioeconomic conditions during the ageing process to prevent type 2 diabetes.

**Contributors** AMP-N, VSSM and ALRV contributed to the conception and design of this study. VSSM and AMP-N were involved in the acquisition of data. MHdS, AMP-N and ALRV contributed to the analysis and interpretation of data. ALRV, AMP-N and LCP were involved in the drafting of the article and ALRV, MP-N, LCP, MHdS and VSSM in revising it for intellectual content. All authors gave final approval of the completed article

**Funding** The São Paulo Foundation for the Support of Research (FAPESP), grant # 04/10524-8.

**Competing interests** None.

**Patient consent** Obtained.

**Ethics approval** The project was approved by the internal review board of CAISM/UNICAMP and was conducted in compliance with the current version of the Declaration of Helsinki and with Resolution 196/96 of the Brazilian National Committee for Ethics in Research (CONEP) and its subsequent revisions.

**Provenance and peer review** Not commissioned; externally peer reviewed.

**Data sharing statement** Extra data can be accessed via the Dryad data repository at http://datadryad.org/ with the doi:10.5061/dryad.nr5j1.

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
