## [Reviewer comments · BMJ Open]

Some articles will have been accepted based in part or entirely on reviews undertaken for other BMJ Group journals. These will be reproduced where possible.

ARTICLE DETAILS

TITLE (PROVISIONAL)	Factors associated with the age of the onset of diabetes in women aged 50 years or more: a population-based study
AUTHORS	Valadares, Ana; Machado, Vanessa; Costa-Paiva, Lúcia; Sousa, Maria Helena; Pinto-Neto, Aarão

VERSION 1 - REVIEW

REVIEWER	Mary Korytkowski University of Pittsburgh Department of Medicine US
REVIEW RETURNED	01-Apr-2014

GENERAL COMMENTS	This paper presents data from a population survey conducted among randomly selected women age > 49 with the purpose of identifying factors associated with risk of being given a diagnosis of diabetes. This is primarily a descriptive study of the population of Campina, Brazil. The sampling procedures are carefully performed. The results identify > 22% of the respondents as having diabetes, with self-reported good health being the primary predictor of remaining free of diabetes. Living with \geq people and BMI at age 20-30 were associated with highest risk for a diagnosis of diabetes. Specific comments: A standardized survey is used for all participants. The survey includes questions about physical activity, smoking, and alcohol use, but results of these analyses are not included or discussed. It would be surprising to have no observed association of physical activity with diabetes risk, and this is worthy of discussion. Table 1 It would be useful to add in 2 additional columns – 1 including only those women who did not report diabetes and another with those reporting diabetes. No questions are asked about other health problems, including depression, which can have a major impact on risk for development of diabetes in women. No information is provided about dietary intake or type of foods consumed which can also impact risk for diabetes. BMI at age 20-30 is capped at ≥ 25 kg/m², yet the discussion provides supporting data for the association of BMI at younger ages being with BMI ≥ 30 kg/m². It is not clear why this one category was used for younger age and two different levels of obesity were used for the older women. Figure 1 carries out the timeline for risk for a diagnosis of diabetes to > 90 years of age, which almost implies some survival benefit of developing diabetes. While the case is made for the sensitivity of using self-reported diabetes, there are other degrees of abnormal glucose tolerance
---

	(e.g. IGT) that may have altered the results of this study. In addition, the question is posed in a way as to whether they were told by a doctor that they had diabetes. Was there any verification that participants had been seen and evaluated for a doctor's appointment in the recent past? There were no questions as to whether these women may have experienced gestational diabetes, or the number of pregnancies which may also impact diabetes risk. Page 11 Line 48 The fact that there was no association between menopausal status and risk for diabetes may be due to the fact that the majority of women were already post-menopausal at the time the study was conducted. Living with more than one additional person was found to be associated with diabetes in this study. On page 12, line 51, this is assumed to be a marker of socioeconomic conditions, yet the data for SEC are not provided (according to monthly income).
--	--

REVIEWER	Dr.Shilpa Paranjape CSIR-National Environmental Engineering Research Institute, Nagpur, INDIA
REVIEW RETURNED	01-May-2014

GENERAL COMMENTS	The study design is acceptable except the fact that correlation of self related health doesn't seem to be appropriate as a causative factor but it could be effect of presence of diabetes. Instead if seperate lifestyle factors like amount of caloric intake per day or extent and tpe of eercises undertaken would have given a better causative etiology to the onset of Type 2 Diabetes Mellitus.Hence this factor , according to my opinion does not give us clear etiological correlation in the above mentioned study. Also it would have been helpful if we knew the hhealth staus that they are mentioning was prior to onset or diagnosis of Diabetes or their current health status.The study limitations only have a brief mention. In my opinion what impact the limitation has on results of the study should be elaborated.
---

VERSION 1 – AUTHOR RESPONSE

This paper presents data from a population survey conducted among randomly selected women age > 49 with the purpose of identifying factors associated with risk of being given a diagnosis of diabetes. This is primarily a descriptive study of the population of Campina, Brazil. The sampling procedures are carefully performed. The results identify > 22% of the respondents as having diabetes, with self-reported good health being the primary predictor of remaining free of diabetes. Living with ≥ people and BMI at age 20-30 were associated with highest risk for a diagnosis of diabetes.

Specific comments:

A standardized survey is used for all participants. The survey includes questions about physical activity, smoking, and alcohol use, but results of these analyses are not included or discussed. It would be surprising to have no observed association of physical activity with diabetes risk, and this is worthy of discussion.

Table 1 It would be useful to add in 2 additional columns – 1 including only those women who did not report diabetes and another with those reporting diabetes.

We have modified Table 1, as suggested.

No questions are asked about other health problems, including depression, which can have a major impact on risk for development of diabetes in women. No information is provided about dietary intake

or type of foods consumed which can also impact risk for diabetes.

We have mentioned now this limitation: "We also were not able to consider all factors that can impact on risks related to diabetes like health problems, gestational diabetes and dietary intake or type of foods consumed."

BMI at age 20-30 is capped at ≥ 25 kg/m², yet the discussion provides supporting data for the association of BMI at younger ages being with BMI ≥ 30 kg/m². It is not clear why this one category was used for younger age and two different levels of obesity were used for the older women.

We have corrected it and have used the same levels of obesity for both younger age and older women.

We have added:

Studies showed that being obese or overweight in a younger age may increase the risk of developing diabetes.

Figure 1 carries out the timeline for risk for a diagnosis of diabetes to > 90 years of age, which almost implies some survival benefit of developing diabetes.

The average age of onset of the disease was 56 years (SD = 11.2) and median 55 years, but it is important to note that up to 92 years (maximum age) just 44% of the sample of 617 women had had diabetes. In short, the accumulated rate of continuation without diabetes at the age of 92 years was 56%.

While the case is made for the sensitivity of using self-reported diabetes, there are other degrees of abnormal glucose tolerance (e.g. IGT) that may have altered the results of this study. In addition, the question is posed in a way as to whether they were told by a doctor that they had diabetes. Was there any verification that participants had been seen and evaluated for a doctor's appointment in the recent past?

No. This study was based on self-report.

There were no questions as to whether these women may have experienced gestational diabetes, or the number of pregnancies which may also impact diabetes risk.

We agree and we have mentioned now this limitation:

It was not possible to consider all factors that can impact on risks related to diabetes like health problems, gestational diabetes and dietary intake or type of foods consumed.

Page 11 Line 48 The fact that there was no association between menopausal status and risk for diabetes may be due to the fact that the majority of women were already post-menopausal at the time the study was conducted.

We agree with reviewer's opinion. The average age of onset of the disease was 56 years (SD = 11.2) and median 55 years. We have now added "The absence of association between menopausal status and risk for diabetes may be due the majority of women were already post-menopausal at the time of onset of diabetes".

Living with more than one additional person was found to be associated with diabetes in this study.

On page 12, line 51, this is assumed to be a marker of socioeconomic conditions, yet the data for SEC are not provided (according to monthly income).

We have corrected the text and added that it was one hypothesis:

"In addition, we may also hypothesize that they may have lower incomes and poorer health conditions"

Reviewer: 2

Reviewer Name Dr.Shilpa Paranjape

Institution and Country CSIR-National Environmental Engineering Research Institute, Nagpur, INDIA

Please state any competing interests or state 'None declared': none declared

We have written at the end of the paper:

Conflicts of interest statement

As corresponding author, I confirm that I have collected ICMJE Uniform Disclosure Forms for Potential Conflicts of Interest from every author and no Conflicts of Interest exist for any of the authors.

The study design is acceptable except the fact that correlation of self related health doesn't seem to be appropriate as a causative factor but it could be effect of presence of diabetes. Instead if separate lifestyle factors like amount of caloric intake per day or extent and tpe of eercises undertaken would have given a better causative etiology to the onset of Type 2 Diabetes Mellitus.Hence this factor , according to my opinion does not give us clear etiological correlation in the above mentioned study. Also it would have been helpful if we knew the hhealth staus that they are mentioning was prior to onset or diagnosis of Diabetes or their current health status.

As it was a cross-sectional study, one can only say that there was an association of onset of diabetes with self-rated health. So, it was not possible to demonstrate that a poor self-rated health was a causative factor or the effect of the onset of diabetes, due to the design of this study.

The study limitations only have a brief mention. In my opinion what impact the limitation has on results of the study should be elaborated.

We have improved our discussion taking into account your suggestions.

Paragraphs:

“As it was a cross-sectional study, one can only say that there was an association of onset of diabetes with self-rated health. So, it was not possible to demonstrate that a poor self-rated health was a causative factor or the effect of the onset of diabetes, due to the design of this study.” Discussion Paragraph 3

“It was not possible to consider all factors that can impact on risks related to diabetes like health problems, gestational diabetes and dietary intake or type of foods consumed.” Discussion Paragraph 6

“Furthermore, in the present study the age of the occurrence of diabetes was also based on the report of the diagnosis made by the physician and other degrees of abnormal glucose tolerance were not taken into account.” Discussion Paragraph 7